# Recent Methods for Modifying Mechanical Properties of Tissue-Engineered Scaffolds for Clinical Applications

**DOI:** 10.3390/biomimetics8020205

**Published:** 2023-05-16

**Authors:** Andrew Johnston, Anthony Callanan

**Affiliations:** Institute for Bioengineering, School of Engineering, University of Edinburgh, Edinburgh EH9 3DW, UK; s1658384@ed.ac.uk

**Keywords:** scaffold, biomimetic, tissue engineering, 3D printing, electrospinning

## Abstract

The limited regenerative capacity of the human body, in conjunction with a shortage of healthy autologous tissue, has created an urgent need for alternative grafting materials. A potential solution is a tissue-engineered graft, a construct which supports and integrates with host tissue. One of the key challenges in fabricating a tissue-engineered graft is achieving mechanical compatibility with the graft site; a disparity in these properties can shape the behaviour of the surrounding native tissue, contributing to the likelihood of graft failure. The purpose of this review is to examine the means by which researchers have altered the mechanical properties of tissue-engineered constructs via hybrid material usage, multi-layer scaffold designs, and surface modifications. A subset of these studies which has investigated the function of their constructs in vivo is also presented, followed by an examination of various tissue-engineered designs which have been clinically translated.

## 1. Introduction

The means of restoring mechanical functionality to damaged biological tissues has proven to be a longstanding challenge to medical practitioners. The complexity of the task is largely due to the intricacy of the native tissue, where biomechanical properties are largely dictated by the extracellular matrix (ECM) [1,2]. This structure consists of a multifaceted network of various constituents such as water, polysaccharides, and proteins such as collagen and elastin [3]. When under load, the ECM relies primarily on these proteins working in conjunction with one another for support [4], with the collagen providing mechanical resistance to deformation under increased load conditions, and elastin being responsible for maintaining the elasticity of the tissue and resistance under relatively low loading [5]. Mechanical integrity and elasticity are not only critical for explicit functions such as joint articulation and other musculoskeletal interactions; they are just as significant for various parasympathetic processes such as peristalsis, respiration, and vasodilation [6,7,8]. The degradation of tissue biomechanics, whether via natural processes such as ageing, or via exposure to circumstances leading to internal or external injury of soft and hard tissues, can precede issues ranging from loss of mobility and discomfort, to critical conditions such as pulmonary fibrosis and supravalvular aortic stenosis [9,10,11].

The complexity of repairing extensively damaged tissues is such that contemporary regenerative medicinal practices are limited in treatment options [12]. The transplantation of fresh tissue is seen as the optimal solution, as the new material is theoretically able to fulfil the physiological and mechanical requirements of the original tissue [13]. However, both allografts and autologous grafts bear several limitations. In the case of allografting, graft-versus-host disease, transplant rejection, bleeding, and infection are constant risk factors [14], whereas autologous grafting is not always an option, due to either previous harvesting of the donor site or systemic disease rendering the tissue unsuitable [15]. An alternative therapeutic in promoting the repair of damaged tissues, stem cell treatment, is largely in its infancy as a research area, and faces several significant hurdles, namely, a high risk of immune response, scalability, and overcoming the negative public perception of such a treatment [16,17]. An alternative means of restoring the mechanical properties of biological tissue is therefore necessary in order to address the ever-increasing demand for biological tissue transplantation.

Tissue engineering is a rapidly expanding field which aims to design constructs which supplement or replace damaged biological tissue. The fabrication of a tissue-engineered graft can be undertaken in a variety of ways, including phase separation, bioprinting, gas foaming and electrospinning techniques [18,19,20,21,22,23]. From these methods, a dense interconnected network of material may be formed, which is termed a scaffold [24]. The purpose of a scaffold is to provide a chemically and mechanically appropriate environment to facilitate cellular growth, which may then be implanted into a given patient [25,26,27]. The ideal tissue-engineered scaffold would allow the development and restoration of fully functional tissue at the graft site, maintaining structural integrity under physiological stress, and ultimately be removed by the body’s natural processes after fulfilling its function [28,29,30].

A key challenge within this area is ensuring the greatest possible resemblance between the mechanical response of the tissue-engineered graft and the host’s native tissue. This is a crucial aspect of a successful design, as biological tissues are load-bearing by nature, regardless of function, from the strongest regions of cortical bone with mechanical strength in the gigapascal range [31] to the most delicate of neural fibres throughout the nervous system whose strength generally lies within the low kilopascal scale [32]. The requirement to maintain the physical integrity of the scaffold while also retaining biocompatibility has given rise to so-called ‘biomaterials’, which the majority of tissue-engineered scaffolds are composed of and which fall into several broad camps: natural polymers, synthetic polymers, ceramics, decellularized matrices, hydrogels, and metals [33,34,35,36,37,38]. Natural polymers can include agarose, alginates, and chitosan while synthetic polymers include polycaprolactone, poly(l-lactic acid), and Poly(ethylene glycol) diacrylate. Ceramics can comprise aluminium oxide, bioglass, and hydroxyapatite, while decellurised matrices can be derived from almost any bodily feature, such as bones and organs. Hydrogels are typically composed of naturally derived constituents such as collagen, gelatine and hyaluronic acid, and metals used for scaffolds can include magnesium, tantalum, and titanium [26,36,39,40,41,42,43,44,45,46,47,48,49,50,51,52,53]. The broad range of characteristics of these materials allows for a vast range of tissue types to be mechanically accounted for; an overview of the typical strength of these material groups in comparison to the stiffness of organic tissues is given in Figure 1 [32,54,55,56,57,58,59,60,61,62,63,64,65,66,67].

One can observe from Figure 1 that particular material types are more suited for certain regions than others; it is clear, for example, that hydrogel alone is an unsuitable replacement for bone tissue. While the consequences of a graft which features an insufficient load-bearing capacity may appear self-evident, the response of surrounding native tissue is often less explicit. For example, a disparity in elastic properties between a tissue-engineered vascular graft and the surrounding native vessel tissue may cause the graft to ineffectively constrict and dilate in tandem with the surrounding vasculature, leading to an increase in smooth muscle cell proliferation, subsequent thrombus formation, and occlusion of the vessel [68,69]. Additionally, when engineered tissue is employed elsewhere in the body, such as for bone or the anterior cruciate ligament, the surrounding tissue will undergo remodelling in accordance with Wolff’s Law and its corollary, Davis’ Law, which state that both osseous and soft tissue, respectively, will structurally adapt themselves in accordance with the mechanical stresses which they encounter [70,71]. This translates to an overcompensation of the native tissue in the case of an insufficiently robust implant, while a construct which exceeds the mechanical requirements of the native tissue can lead to the degradation of the surrounding architecture, with both scenarios compromising the functionality of the original biological tissue [72,73]. It is therefore crucial that the mechanical properties of engineered tissue are characterised in such a way as to best compliment the surrounding biomechanical environment of the native tissue. This does not necessarily require achieving perfect parity between the stiffness of the existing tissue and a given implant; indeed, several of the studies considered in this review suggest that sufficiently mechanically resilient biocompatible materials are perhaps more suitable for the promotion of tissue regeneration than those who endeavour to match the native tissue’s properties in a mechanical context alone. Nevertheless, ensuring a high degree of equivalence between an engineered and a given native tissue’s mechanical properties is a functional requirement of a successful engineered graft.

To address this challenge, tissue engineering researchers are investigating a broad range of solutions to optimise the mechanical response of tissue-engineered grafts, while also maintaining the biocompatibility of the design. The current studies in this field include the use of multiple materials in conjunction with one another, such as polycaprolactone and collagen, bi- and multi-layered scaffolds which often delegate the functional requirements of the construct to each layer, and modifying existing scaffold parameters such that their performance may be tuned to better suit the conditions of the native tissue [74,75,76]. Success in these areas has led to multiple studies in vivo with such designs, and, in some cases, to full clinical translation, whereby the cases of which are examined in Section 7 of this work.

To support the development of these designs, a broad range of literature is available which describes the mechanical properties of specific native tissue types and engineered tissue scaffolds. These include bone, cartilage, liver, kidney, osteochondral tissue, muscle, and tendon, and utilise a broad range of mechanical property assessment methods, including compressive, dynamic, nanoindentation, shear, and tensile testing, with additional methods including finite element analysis (linear and non-linear) and ultrasound [58,77,78,79,80,81,82]. Using these techniques, native tissue properties such as Young’s modulus, yield stress, failure strength and strain, fracture toughness, and viscoelastic behaviour can be assessed. Review articles of this nature are invaluable as resources for the engineering of specific tissue types. However, a comparative examination of a broad variety of tissues, materials, fabrication, and testing methods would offer a unique insight into the effects of parameters and their effect on the mechanical performance of a particular design, which could potentially be applied to other tissue types. A review of this nature, to the best of the authors’ knowledge, has not been presented in the literature before.

This work provides a review of the current techniques by which researchers have successfully modified the mechanical properties of tissue-engineered grafts. The scope of this review will encapsulate all applicable construction methods which are currently employed in this regard, and all tissue types will be considered in accordance with the focus of each study. A Appendix A, provides a greater deal of insight into the studies considered in this work, with a brief summary of each publication provided therein. The extent of this review is all relevant work published within the years of 2021 to 2023, and it encompasses primary articles reporting on the use of hybrid-, multi-layer-, and surface-modified-type scaffolds, as well as clinically translated designs whose use and outcomes were published within the above chronological range. Articles published prior to these dates were utilised as citations for statements and empirical data where appropriate.

## 2. Mechanical Properties of Biological Tissues and Their Influence

The mechanical properties of biological materials are a product of a complex array of proteins, cells, and interstitial fluid flow, whose interactions characterise the response of their respective tissue under load [83,84]. As a product of their intricacy, biological tissues exhibit complex mechanical properties such as heterogeneity, viscoelasticity, and anisotropy [83,84,85]. Accurately assessing the mechanical properties of these tissues is therefore intrinsically challenging, with research groups often attaining a variety of results for the same tissue. Table 1 illustrates this, featuring a short collection of various biological tissues, and a generally broad range of respective mechanical properties, encountered by several research groups for the same mode of analysis.

The diversity in these results can not only be attributed to the structural composition of the tissues alone; factors such as hydration and age are also known to affect the mechanical response of biological tissues [104,105]. This process of tissue-specific cells responding to mechanical stimuli is termed mechanotransduction, which is a means by which organic tissue may convert external loading, such as tension or vibrational waves, to biochemical, biophysical, or molecular signals via a series of events culminating in a cellular response [106]. Figure 2 illustrates this process, beginning with a given external loading event encountered by the body’s mechanosensors. This is termed mechanocoupling, and, depending on the load type, is translated by one or more mechanosensors to adjacent cells in the ECM [107]. Via a complex series of intra-cellular interactions, these mechanical signals are converted to various signal types which instruct the relevant cell groups to behave in a particular way; for example, an increase in mechanical load due to extensive physical activity can lead to compensatory growth in relevant skeletal and muscular members [106,108].

It is the cellular response to various conditions which allows the human body to maintain homeostasis, while also quickly adapting to environmental circumstances; vasoconstriction and vasodilation are common examples of this [109]. However, a crucial caveat is that biological tissues have evolved to expect a certain degree of physiological loading. A reduction in the stress and strain forces imparted on, for example, the head region of the femur as a result of a hip implant, can in turn lead to the ‘stress-shielding’ effect [110]. This is where a non-native element acts as the main load-bearing facilitator during typical movement (e.g., walking or running). As a result, the incentive for cells to remodel and strengthen their respective tissues during mechanotransduction is lowered and local bone density is reduced, potentially leading to implant loosening, stress fracturing, and abnormal bone development [111,112]. It follows that other tissue-engineered designs are bound by the same factors; a mechanically under- or over-strengthened graft is not only impractical, but can cause deleterious effects on neighbouring local tissue groups as outlined above. This highlights the importance of mechanical parameters when designing a tissue-engineered graft.

## 3. Hybrid Materials

The non-linear biomechanical response of a tissue under load is difficult to reproduce in synthetic materials alone, as this response is characterised by complex biological structures and interactions, as described in Section 2. Current research indicates that a hybrid polymer blend, consisting of both synthetic and natural materials, allows for constructs which better represent this intricate behaviour [113]. This is due to the naturally derived constituent typically retaining its microstructural details, which can promote cell attachment and integration; this is beneficial when employing a scaffold in a biological environment [114,115]. A broad range of hybrid polymer combinations are currently undergoing extensive research, ranging from binary PCL-hydroxyapatite designs [29] to complex three- or four-part hybrid amalgamations [116,117]. Table 2 provides an overview of a selection of such work, including the materials and fabrication methods employed during each study, the intended tissue type, and which mechanical properties were assessed throughout the work. As an extension of this, Table 3 explores the Young’s moduli value ranges attained in each of the studies in question, quantifying the effects of these material combinations.

The scope for hybrid biomaterial design is vast, which allows researchers to consider a broad range of materials. For example, several research groups have recently investigated the inclusion of multi-walled carbon nanotubes (MWCNTs), which are nanometre-diameter carbon-based tubular constructs, with the aim of utilising the nanotubes’ high tensile strength and electrical/thermal conductivity to enhance the overall biomechanical properties of their scaffold. Sang et al. dispersed MWCNTs with concentrations ranging between 1, 3, and 5% through a chitosan/polyethylene glycol scaffold, which was prepared by freeze-drying, for the purpose of neural tissue engineering [127]. The electrical conductivity of neurons is key to their function, and one of the key findings of this analysis was the rise in electrical conductivity of the scaffold in proportion to increasing MWCNT concentrations. Interestingly, the elastic modulus of the hybrid scaffold was also enhanced with an increasing concentration of MWCNTs, from 1.17 kPa to 2.09 kPa, demonstrating the load-bearing abilities of carbon nanotubes. As another example of this, Ramasamy et al. employed MWCNTs for use in a poly(l-lactic acid)–boron nitride piezoelectric nanosheet hybrid, where it was found that the combination of these materials led to a significantly stronger graft when compared to unmodified poly(l-lactic acid) nanofibers quantified by a 666% increase in tensile strength in contrast [128].

Hybrid plant-based designs are also promising in this area; a recent study by Jiawei et al. investigated the inclusion of hydroxyapatite into sugarcane stem, whose lignin had been removed (i.e., delignified) via soaking in a NaClO_2_/acetate buffer solution. The degree of delignification induced a proportionate increase in the sugarcane stem porosity, albeit at the cost mechanical performance under compressive load, with the greatest extent—8 h of immersion—reducing the compressive modulus of 14.6 MPa for pure sugarcane stem to 4.82 MPa. An optimal soaking time of 4 h was established, which promoted an elongated elastic response from the scaffold under load whilst maintaining structural integrity [129].

## 4. Multi-Layer Scaffolds

The challenge of tuning the elastic response of a scaffold in tandem with balancing its biocompatibility characteristics has led several research groups to fabricate scaffolds consisting of multiple layers. This approach allows for the functionalisation of each stratum independently and more closely represents the layered nature of most biological tissue, such as arterial walls, osseous tissue, and oesophageal tissue [130,131,132]. The potential impact of various material and fabrication combinations has allowed researchers to explore a broad range of designs consisting of materials such as silk, graphene, and titanium [133,134,135]. Table 4 provides a brief overview of the groups which have investigated the potential of bi- and multi-layered scaffold designs, while Table 5 illustrates the Young’s modulus ranges attained during these studies.

While the properties of multi-layered scaffolds are largely determined by the material selection, several groups have investigated the potential of applying novel fabrication methods in order to construct a multi-layered construct which can function within the physiological load range. Killian et al. studied the combination of calcium phosphate cement (CPC), fabricated via 3D printing, and PCL, drawn using a melting electrowriting (MEW) process for the purpose of bone tissue engineering [147]. These two materials were printed upon one another up to 10 layers in height, creating a grid-like pattern of CPC which contained multiple interspatial variations, and from which the strands of PCL fibre could intersect. Interestingly, while the addition of the PCL fibres slightly diminished the yield strength and elastic modulus of the scaffolds (which were approximately 4–10 and 40–85 MPa, respectively), the inclusion of these fibres would occasionally cause the construct to mechanically yield twice under load: once for the PCL strands, and again as the CPC failed, offering an element of redundancy to the design. Liu et al. fabricated a bilayered construct by combining electrospinning and 3D printing processes, with the intention of promoting guided bone regeneration [148]. The electrospun layer consisted of a PCL/gelatine membrane, while the 3D-printed scaffold consisted of a PCL/gelatine/HA mixture, which were combined by dissolving the electrospun membrane and adding it to the scaffold. It was found that the compressive strength of the resulting scaffold, at 13.86 MPa, closely resembled that of cancellous bone (up to ~13 MPa [149]). The similarity of these compressive stress values suggests that a greatly diminished stress-shielding effect would be observed in clinical trials, in comparison to existing bone implant devices formed from a Ti-6Al-4V alloy with a Young’s modulus of 110 GPa [150].

In addition, Thompson et al. employed a process known as embedded 3D printing to fabricate a multi-layer scaffold in an effort to replicate human vocal cord tissue [151]. This method utilises a cavity which has pre-printed ink filaments embedded within a given medium, allowing for the fabrication of a component without interference from gravity or requiring additional supports (Figure 3) [152]. For this particular study, various silicone elastomers and thinners were used to construct each layer, and then they were cured via a 700 W microwave. The elastic modulus varied depending on the layer assessed, with the superficial lamina propria (SLP) providing a 0.91 kPa tensile modulus, and the epithelium increasing this value to 39.74 kPa, which lies within the range of human vocal cords, at approximately 30 kPa [153]. The group were also successful in functionalising the construct by designing it to exhibit flow-induced vibration, representing human phonation.

## 5. Surface Modification

The mechanical properties of engineered tissue are not necessarily set in stone. By design, a substance such as a polymeric compound may have its constituents altered such that it may better serve its intended function [154,155]. In a similar manner, a tissue-engineered graft, being typically composed of one or more complex substances, may be modified in such a way that a desired property, such as increased mechanical strength, can be achieved without having to resort to less suitable or more expensive material alternatives, via processes such as annealing or salt leaching [156,157]. Recent studies have considered a broad range of surface modifications, depending on the desired functionality of the design. These range from topographical nano-modification of the scaffold [158] to grafting post-fabrication and polymer coatings [159,160]. Table 6 offers an overview of the materials, fabrication methods, and modification methods employed in recent studies involving surface modification of tissue-engineered grafts. Table 7 expands on this by showing the Young’s modulus value ranges achieved during each of the studies featured in this section.

Surface modification also has the capacity to facilitate the functionalisation of more unlikely material candidates for this research into more suitable substances. An example of this is offered by Mahendiran et al., who examined the potential of cellulose scaffolds derived from *Borassus flabellifer* for the purpose of bone tissue regeneration [166]. The scaffolds were derived from the plant’s immature endosperm which was then, after washing and oxidising the scaffolds, modified by two organosilanes; amino-terminated aminopropyltriethoxysilane (APTES) and methyl-terminated octadecyltrichlorosilane (OTS). The subsequent scaffolds formed a foamy architecture, and, in comparison to the unmodified constructs employed as a control during the study, both the APTES and OTS treatments enhanced the compressive modulus of the material, from approximately 0.3 MPa to 0.9 MPa and 1.2 MPa, respectively. Crucially, both treatment methods also bore osteoinductive properties, demonstrating the potential of such a design in a clinical context.

Nanotopographical roughness is a feature which may also be employed to modify the properties of a given scaffold. The influence of surface roughness on cell behaviour is well documented, with several groups reporting relative increases in cell proliferation, adhesion, and desired protein expression when seeded on irregular scaffold surface morphologies [167,168,169]. One method of creating roughness on engineered scaffolds is alkaline hydrolysis, which was employed by Meng et al. to alter an existing PLLA design for the purposes of bone tissue engineering [170]. For this particular study, the scaffolds, formed via MEW, were immersed in alkali solutions consisting of concentrations of 0.25 M and 0.5 M sodium hydroxide (NaOH) in ethanol (1:1 ratio) for 1, 2, 3, and 4 h at a time. While the effect of this process was perhaps most notable in terms of cell count, which, after 15 days, was largely increased in comparison to the control scaffold, perhaps the most surprising result was the improved tensile modulus of the 0.5 M NaOH scaffold. After 1, 2, and 3 h periods of immersion, a maximum modulus of 5 GPa was achieved, in comparison to 2.5 GPa of the control scaffold. It was theorised that the alkaline treatment process increased the crystallinity of the individual PLLA fibres, thus providing additional tensile strength for the scaffold overall.

## 6. In Vitro Limitations and Animal Research

The mechanical and biological performance of a tissue-engineered scaffold can be analysed in a number of ways: mechanically, via the methods described in Section 1 such as compressive, shear, and dynamic testing [61,171,172], while biological viability of the scaffold can be studied in vitro via cell seeding of primary or clonal cell lines, from which parameters such as cell adhesion, differentiation, and viability can be derived [173,174]. While these methods may suggest how such a design would perform in a clinical context, there are several clear limitations to such an evaluation, which exist primarily due to the complexity of physiological in vivo conditions. Mechanically, the human body is in a constant state of flux; internal and external loads are met with biomechanical responses as a product of the mechanotransducive process outlined in Section 2. As an illustration of these mechanical stresses, Figure 4 features an arterial section cutaway diagram, where the biological tissue is subjected to physiological stress conditions as a result of hemodynamic flow.

A construct designed to operate in such an environment, such as a vascular graft, would be expected to function under these load conditions; mechanical testing is therefore key, prior to the further development of such a design. However, classical mechanical assessment methods such as those outlined earlier in this section, while perhaps indicative of a particular design’s fundamental mechanical properties, are often incapable of evaluating the graft’s performance under true physiological conditions, as these are typically non-linear, heterogeneous loads which are nearly impossible to replicate via contemporary testing equipment [175].

The complex nature of these forces has led scientists to explore several avenues of research in an effort to better comprehend, and design for, their impact on tissue-engineered designs. Recent advances in finite element analysis (FEA) and computational fluid dynamics (CFD) analyses have allowed researchers to model tissues such as pulmonary arteries [176], airways [177], ventricles [178], and eye lenses [179], which carries benefits ranging from a greater understanding of the morphological and physiological characteristics of these tissue types, to assisting surgeons during complex surgical procedures [180,181]. A brief overview of additional studies such as these is given in Table 8.

Despite the promising outlook of this research, a true estimation of the in vivo performance of a tissue-engineered scaffold is still beyond the grasp of contemporary computational or in vitro methods.

The next logical step, therefore, is to assess the viability of tissue-engineered implants in vivo. This will allow researchers to understand how viable their scaffold’s design is once implanted into biological tissue, and will also validate the use of particular materials and construction methods within living organisms. Table 9 subcategorises the previously described studies thus far into those which assessed the performance of their designs in an in vivo setting, and provides an overview of the study parameters, as well as a brief summary of each respective group’s findings.

The assessment of tissue-engineered designs in vivo can also offer insight into cell behaviour when uncommon scaffold morphologies are implanted within the animal. Feng et al. examined the effects of implanting a conch-like scaffold, featuring a helical inner structure, into the upper femoral region of New Zealand rabbits, with the aim of assessing such a design to promote guided bone regeneration [190]. Formed via 3D printing with β-tricalcium phosphate, various diameters, pitches, and pore sizes of the scaffold were examined, with a maximum compressive modulus of 1.75 GPa achieved with a 9.8 mm scaffold diameter, 1 mm pore diameter, and 2.4 mm pitch. It was found that the distinctive design of the scaffold encouraged a capillary action effect when placed within a cell media solution; this phenomenon was fully demonstrated when the scaffold was assessed in vivo, as the cells rapidly proliferated up the helical section, illustrating the directional osteoinductive benefits of such a scaffold morphology.

Despite developments in this area, in vivo testing in animals, especially in the current exploratory phase of tissue engineering research, suffers from a widely variable animal-to-human translational success rate [191]. This is primarily due to differing physiology between animals and humans, which also extends to genetics and epigenetics, as well as the low reproducibility of such experiments and lack of use of prospective systematic reviews [191,192]. As a potential solution to this, a rapidly emerging technology termed organ-on-chip removes the in vivo aspect of organic tissue development by providing a biomimetic substrate upon which living tissue can develop. This tissue can progress to function as an organoid, with models such as liver- and kidney-on-chip providing valuable insights as to how such tissue may be fully vascularised and function prior to use in vivo [193,194]. A brief overview of which organ-on-chip models are currently in use is provided in Table 10.

Organ-on-chip technology bears significant promise, as it may provide a more agile and less ethically challenging means of assessing tissue-engineered constructs. However, one can observe from Table 10 that consideration of the mechanical properties of the engineered tissue examined in these designs is often not assessed in these designs.

## 7. Clinical Translation, Challenges, and Future Outlook

The use of synthetic designs to supplant existing damaged native tissue is a long-established practice, with the earliest total hip arthroplasty, made with ivory, being recorded in Germany in 1891 [210]. Scientific advancement since then has led to several crucial developments in the field of tissue engineering, such as the creation of various polymers, the advents of 3D printing and electrospinning, and a broader understanding of the biomechanical underpinnings of the human body. However, as illustrated in this article thus far, tissue engineering is still a developing field. Further progression in key areas such as the mechanical characteristics, biocompatibility, and fabrication methods involved in proposed designs are required before the use of tissue-engineered constructs in clinical settings is likely to become commonplace.

A further hurdle is that tissue-engineered designs are subject to various regulatory frameworks depending on the jurisdiction in which the design is to be marketed and sold within. While several jurisdictions in the European Union, United States of America, Canada, Australia, Japan, and South Korea have various classifications for tissue-based products, many others do not [211]; difficulties in introducing or adapting legislation for such complex products will likely involve protracted legal processes, potentially limiting the use of tissue-engineered designs within these regions for a significant period.

Regardless, progress in this field is clear, with several novel tissue-engineered implants having recently been successfully trialled. Table 11 provides an overview of these trials, including the material type, fabrication method, intended tissue type or region, the procedure method and study length, total study size, and a brief summary of the results of each trial.

A clear trend from Table 11 is that, by and large, 3D printing techniques have been the dominant fabrication method for the fabrication of clinically translated implants thus far. The logic for this is relatively self-evident: human soft tissue and cardiovascular and musculoskeletal systems are unique as a result of age, health, and genetics; injuries or pathologies pertaining to these systems increase this distinctiveness further, often requiring an equally bespoke treatment method that a technique such as additive manufacturing is readily able to provide. Additionally, the short lead times associated with 3D printing as a result of rapid prototyping, manufacturing, and delivery times are a boon to often time-sensitive clinical circumstances [224].

Whilst the continuous utilisation of 3D printing technology as the basis for the clinical translation of tissue-engineered designs remains to be seen, alternative fabrication methods such as electrospun grafts, hydrogel systems, and bioprinting continue to be developed in parallel, bringing tissue-engineered constructs ever closer to clinical and commercial viability.

## 8. Conclusions

The limitations of contemporary surgical and therapeutic techniques to restore the original functionality of damaged organic tissue have driven the development of various tissue-engineered platforms, which aim to support and encourage the regeneration of healthy autologous tissue. One of the key challenges in this area is establishing a high degree of similarity in the mechanical properties between the engineered scaffold and the surrounding biological tissue. This is due to the typical inability of manufactured materials to replicate the complex biomechanics of organic tissue. This capability, however, is crucial for a successful tissue-engineered design.

In response to this issue, current tissue engineering research is examining a broad spectrum of potential designs aimed at enhancing the mechanical and biological compatibility of tissue-engineered constructs. From the studies described in this work, it is clear that relatively well-documented fabrication methods such as electrospinning and 3D printing can produce designs with, in some cases, a high degree of mechanical equivalence to native tissues. Techniques such as bioprinting and MEW, while promising, will likely require further validation before widespread use is achieved. In terms of materials, the usage of particular polymers such as PCL and PLA, ceramics such as HA and β-TCP, and hydrogels consisting of gelatine and alginate have seen widespread use, while the effects of more novel materials such as CNTs and quantum dots are yet to be fully examined in a tissue engineering context.

A notable observation from these studies is that accounting for the complex range of mechanical properties present in biological tissues is generally beyond the remit of a single-material-based design. Rather, a multifaceted approach consisting of multiple material types, layers, or surface treatment methods working in tandem is likely to yield the most successful designs. In terms of translation to clinical use, the time-sensitive nature of surgical treatment suggests that 3D printing, as a fabrication method, is a strong contender for the successful realisation of these designs. Regardless of approach, it is clear that with further development, tissue engineering is set to act as a transformative treatment method.

## Figures and Tables

**Figure 1 biomimetics-08-00205-f001:**
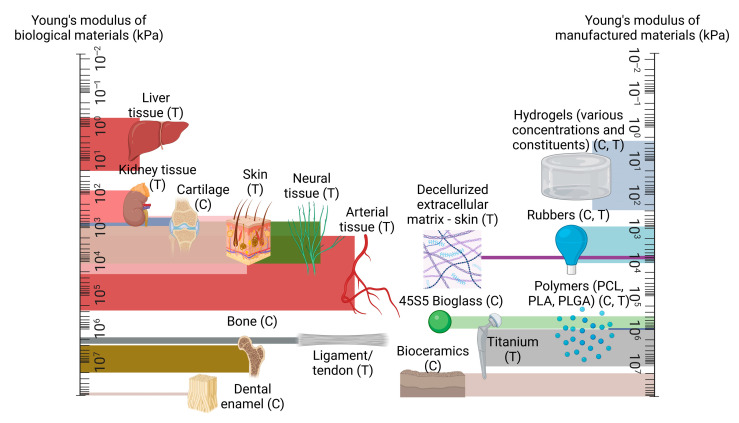
A comparison of mechanical strength between native tissue types, in comparison to manufactured materials, presented on logarithmic scales, with compression and tension testing denoted with (C) and (T) as appropriate. Created using Biorender.com, 15 May 2023.

**Figure 2 biomimetics-08-00205-f002:**
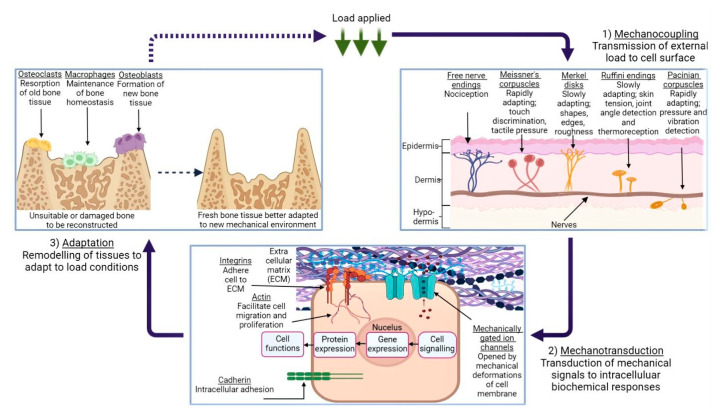
An illustration of the mechanotransduction cycle, in the context of osseous tissue remodelling in response to external loading applied to a given area of skin, via the medium of a generic cuboidal cell. (**1**) demonstrates the various types of mechanoreceptors present in human dermis and their function, while (**2**) depicts the mechanotransduction pathway, followed when the ECM is subject to deformation. (**3**) shows the remodelling process of bone tissue in response to this loading, and the roles which the various cells play in this process. Created using BioRender.com, 24 April 2023.

**Figure 3 biomimetics-08-00205-f003:**
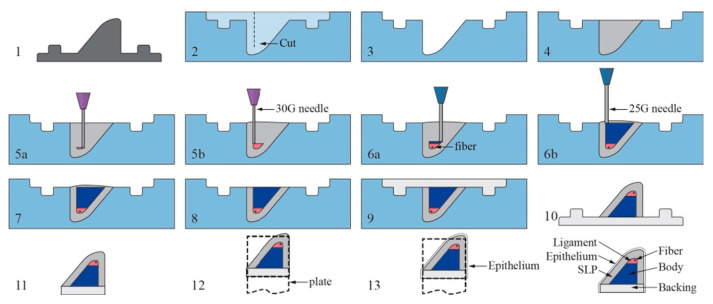
An illustration of the embedded 3D printing process of the vocal cords. Steps as follows: (**1**) Create positive mold of SLP layer. (**2**) Create reservoir with cut for fiber insertion. (**3**) Coat reservoir with release agent. (**4**) Fill reservoir with SLP support matrix silicone. (**5a**,**b**) Print ligament layer within reservoir, beginning near reservoir bottom, and then insert fiber. (**6a**,**b**) Print body layer beginning at interface of ligament layer. (**7**–**8**) Remove overflow material and then cure in microwave. (**9**) Pour backing into remaining cavity and let model cure and cool completely. (**10**) Remove VF model from reservoir. (**11**) Trim excess backing material and gently clean model with acetone. (**12**) Attached model to mounting plate. (**13**) Pour epithelial layer and cure. Reproduced with the publisher’s permission [152].

**Figure 4 biomimetics-08-00205-f004:**
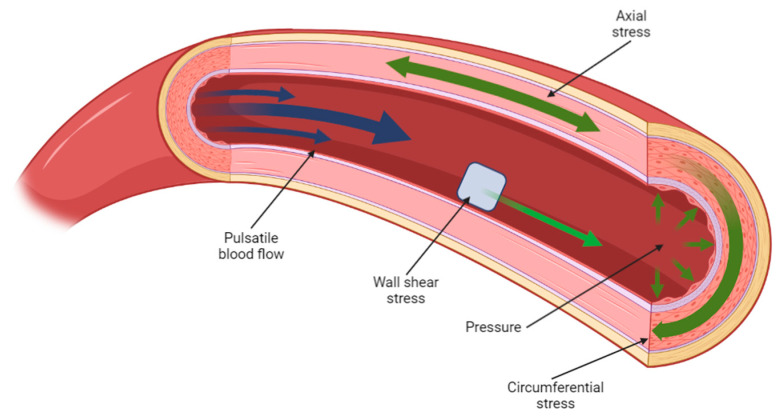
A cross-sectional diagram of arterial flow, illustrating the various load conditions present at any one time in the cardiovascular system. Created using BioRender.com, 6 March 2023.

**Table 1 biomimetics-08-00205-t001:** A table of selected tissue types and their mechanical properties.

Tissue Type	Failure Strain	Yield Stress (MPa)	Young’s Modulus (MPa)	Shear Modulus (MPa)	Reference
Coronary Artery	0.45	0.39–1.8	1.55 ± 0.26	0.3	[86,87,88,89,90]
Cartilage	0.183	4.58 ± 2.04	0.5–0.9	0.26–0.32	[91,92,93,94]
Bone	0.25–0.67	71.56	17,900–19,080	3600	[95,96,97]
Skin	1.5	17–21	60–70	0.002–0.008	[65,98,99]
Spine/ Sciatic/Ulnar Nerve	0.293–0.73	11.7	0.7–10	0.02–0.054	[100,101,102,103]

**Table 2 biomimetics-08-00205-t002:** Summarisation of materials, fabrication methods, intended tissue type, and which mechanical properties were altered and assessed during studies which employed a hybrid polymer blend.

Tissue Type	Material	Fabrication Method	Mechanical Properties Assessed				Author, Year, Reference
			Tensile/Compressive Modulus	Failure Strain	Ultimate Tensile Strength	Storage/Loss Modulus	
Bone	PCL + HA	3D printing	✓	✓			Rezania et al. (2022) [29]
Bone	Gelatine + HA + hPE	3D printing	✓	✓			Lee et al. (2022) [117]
Bone	Col and Col + PGA and Col + PGA/HA + PGA and Col + PGA/HA + PGA (2-layer membrane)	3D printing	✓	✓		✓	Nguyen et al. (2022) [115]
Bone	HA-NFs + GelMA	Hydrogel solution	✓	✓	✓		Wang et al. (2022) [118]
Bone	Gelatine + GO + PHEMA	Freeze-drying	✓				Tabatabaee et al. (2022) [119]
Cartilage	Alginate (Core) + Chitosan (Shell and gel) + SF(gel)	Core-shell microspheres	✓	✓	✓		Min et al. (2022) [120]
Cartilage	GelMA + HyAMA + chondrospheroids	Hydrogel solution	✓				Wang et al. (2022) [118]
Cartilage	PEC + SF + SA + PrP	Phase separation	✓				Singha et al. (2022) [18]
Cartilage	rSF and rSF + rGO	Electrospinning	✓			✓	Dorishettya et al. (2022) [121]
Cartilage/Bone	Gellan gum + Alginate sodium and Gellan gum + Alginate sodium + TMP-BG	3D Printing	✓	✓	✓	✓	Chen et al. (2022) [122]
Kidney	PCL + Laminin + Span80™ emulsion	Electrospinning	✓	✓	✓		Baskapan et al. (2022) [123]
Nerve	PLA + Col	Electrospinning	✓	✓	✓		Xu et al. (2022) [124]
Skin	PCL + SF + SESM + Gelatine and PCL + SF + SESM + MC	Electrospinning	✓	✓	✓		Salehi et al. (2022) [125]
Skin	PCL+ Col and PCL + Col + ZnO and PCL + Col + ZnO + VEGF	Electrospinning	✓	✓	✓		Li et al. (2021) [126]

PCL: Polycaprolactone, HA: Hydroxyapatite, hPE: human placental extracts, PGA: Poly-glutamic acid, Col: collagen, HA-NFs: Hydroxyapatite nanofibers: GelMA: Methacrylated gelatine, GO: graphene oxide, PHEMA: Poly(2-hydroxyethyl methacrylate), TMP-BG: Thixotropic magnesium-phosphate-based gel, SF: silk fibroin, HyAMA: Methacrylated hyaluronic acid, PEC: Polyelectrolyte complexation, SA: sodium alginate, PrP: platelet-rich plasma; rSF: regenerated silk fibroin, SESM: soluble eggshell membrane, MC: Methylcellulose, ZnO: zinc oxide, VEGF: vascular endothelial growth factor, dECM: decellurised extra cellular matrix.

**Table 3 biomimetics-08-00205-t003:** An overview of Young’s modulus value ranges attained during studies presented in Table 2.

Tissue Type	Material	Fabrication Method	Elastic Modulus Range Attained in Tension (T) or Compression (C)						Author, Year, Reference
			<1 kPa	1–100 kPa	100–1000 kPa	1–100 MPa	100–1000 MPa	>1 GPa	
Bone	PCL + HA	3D printing				✓ (C)	✓ (T)		Rezania et al. (2022) [29]
Bone	Gelatine + HA + hPE	3D printing			✓ (C)				Lee et al. (2022) [117]
Bone	Col and Col + PGA and Col + PGA/HA + PGA and Col + PGA/HA + PGA (2-layer membrane)	3D printing		✓ (C)	✓ (C)				Nguyen et al. (2022) [115]
Bone	HA-NFs + GelMA	Hydrogel solution		✓ (C)					Wang et al. (2022) [118]
Bone	Gelatine + GO + PHEMA	Freeze-drying				✓ (C)			Tabatabaee et al. (2022) [119]
Cartilage	Alginate (Core) + Chitosan (Shell and gel) + SF(gel)	Core-shell microspheres				✓ (C)			Min et al. (2022) [120]
Cartilage	GelMA + HyAMA + chondrospheroids	Hydrogel		✓ (C)					Wang et al. (2022) [118]
Cartilage	PEC + SF + SA + PrP	Phase separation			✓ (C)				Singha et al. (2022) [18]
Cartilage	rSF and rSF + rGO	Electrospinning			✓ (C)	✓ (C)			Dorishettya et al. (2022) [121]
Cartilage/Bone	Gellan gum + Alginate sodium and Gellan gum + Alginate sodium + TMP-BG	3D Printing		✓ (C)	✓ (C)	✓ (C)			Chen et al. (2022) [122]
Kidney	PCL + Laminin + Span80™ emulsion	Electrospinning		✓ (T)					Baskapan et al. (2022) [123]
Nerve	PLA + Col	Electrospinning			✓ (T)	✓ (T)			Xu et al. (2022) [124]
Skin	PCL + SF + SESM + Gelatine and PCL + SF + SESM + MC	Electrospinning				✓ (T)			Salehi et al. (2022) [125]
Skin	PCL+ Col and PCL + Col + ZnO and PCL + Col + ZnO + VEGF	Electrospinning			✓ (T)	✓ (T)			Li et al. (2021) [126]

**Table 4 biomimetics-08-00205-t004:** Summarisation of materials per layer, fabrication methods, intended tissue type, and which mechanical properties were altered and assessed during studies which employed a multi-layer scaffold.

Tissue Type	Material per Layer	Fabrication Method	Mechanical Properties Modified and Assessed				Author, Year, Reference
			Tensile/ Compressive Modulus	Failure Strain	Ultimate Tensile Strength	Storage/Loss Modulus	
Bone	PLLA/Cell sheet	Electrospinning	✓	✓			Tevlek et al. (2021) [136]
Cartilage	PCL + Gelatine/Gelatine + Alginate	Electrospinning	✓				Semitela et al. (2021) [137]
Cartilage	Alg + HA 50:50 (~70 layers) and Alg + HA 70:30 (~70 layers)	Bioprinting	✓	✓			Janarthanan et al. (2022) [138]
Osteochondral	Col + PLGA and Ti/Col + PLGA	3D printing/freeze-drying	✓				Yang et al. (2021) [135]
Osteochondral	Ti/PLA/ Col + PLGA and Col + HA	3D printing	✓				Tamaddon et al. (2022) [139]
Osteochondral	PCS/Col + HA	Freeze-drying	✓				Rashidi et al. (2021) [140]
Osteochondral	PCL + PEO + PES and PCL + PEO/rGO + HA-Sr + PES-BH_0.5%_	Electrospinning	✓				Dargoush et al. (2022) [134]
Osteochondral	Silk and BMP-2 + Silk and Silk/Silk (bilayer) and BMP-2 + bilayer and BMP-2 + bilayer/TGF-β1 + SilMA	Hydrogel solution	✓	✓	✓		Wu et al. (2021) [133]
Osteochondral	GelMA + PEO + HA/GelMA + PEO + HA/GelMA + PEO	Hydrogel solution	✓			✓	Li et al. (2022) [141]
Periodontal	PCL/BG and PCL/HyA	3D printing	✓	✓			Nejad et al. (2021) [142]
Vascular	PLGA/PCL	Electrospinning	✓				Bazgir et al. (2021) [143]
Vascular	PCL/PCL	Electrospinning	✓	✓	✓		Li et al. (2021) [144]
Vascular	RHC + PCL/PEO + PCL	Electrospinning	✓	✓	✓		Do et al. (2021) [145]
Vascular	dECM/GAGF/dECM/dECM	Decellurizing and gas foaming	✓	✓	✓		Smith et al. (2022) [146]

Ti: titanium, PLGA: Poly-lactic-co-glycolic acid: PLA: Polylactic acid, PCS: polymerised chondroitin sulphate, PEO: Polyethylene oxide, PES: Polyethersulfone, HA-Sr: Strontium-doped hydroxyapatite, BH: Benzyl hyaluronan, BMP-2: bone morphological protein-2, TGF-β3: transforming growth factor-β3, SilMA: Methacrylated silk, BG: bioglass, HyA: hyaluronic acid, RHC: recombinant human collagen, GAGF: Glycosaminoglycan foam, PLLA: Poly(L-lactic acid).

**Table 5 biomimetics-08-00205-t005:** An overview of Young’s modulus value ranges attained during studies presented in Table 4.

Tissue Type	Material per Layer	Fabrication Method	Elastic Modulus Range Attained in Tension (T) or Compression (C)						Author, Year, Reference
			<1 kPa	1–100 kPa	100–1000 kPa	1–100 MPa	100–1000 MPa	>1 GPa	
Bone	PLLA/Cell sheet	Electrospinning				✓ (T)			Tevlek et al. (2021) [136]
Cartilage	PCL + gelatine/Gelatine + Alginate	Electrospinning			✓ (T)	✓ (T)			Semitela et al. (2021) [137]
Cartilage	Alg + HA 50:50 (~70 layers) and Alg + HA 70:30 (~70 layers)	Bioprinting		✓ (C)					Janarthanan et al. (2022) [138]
Osteochondral	Col + PLGA and Ti/Col + PLGA	3D printing/freeze-drying				✓ (C)			Yang et al. (2021) [135]
Osteochondral	Ti/PLA/ Col + PLGA and Col + HA	3D printing				✓ (C)	✓ (C)	✓ (C)	Tamaddon et al. (2022) [139]
Osteochondral	PCS/Col + HA	Freeze-drying		✓ (C)					Rashidi et al. (2021) [140]
Osteochondral	PCL + PEO + PES and PCL + PEO/rGO + HA-Sr + PES-BH_0.5%_	Electrospinning		✓ (T)					Dargoush et al. (2022) [134]
Osteochondral	Silk and BMP-2 + Silk and silk/silk (bilayer) and BMP-2 + bilayer and BMP-2 + bilayer/TGF-β1 + SilMA	Hydrogel solution			✓ (C)	✓ (C)			Wu et al. (2021) [133]
Osteochondral	GelMA + PEO + HA/GelMA + PEO + HA/GelMA + PEO	Hydrogel solution		✓ (C)					Li et al. (2022) [141]
Periodontal	PCL/BG and PCL/HyA	3D printing			✓ (C)	✓ (C)			Nejad et al. (2021) [142]
Vascular	PLGA/PCL	Electrospinning				✓ (T)			Bazgir et al. (2021) [143]
Vascular	PCL/PCL	Electrospinning				✓ (T)			Li et al. (2021) [144]
Vascular	RHC + PCL/PEO + PCL	Electrospinning				✓ (T)			Do et al. (2021) [145]
Vascular	dECM/GAGF/dECM/dECM	Decellurizing and gas foaming				✓ (T)			Smith et al. (2022) [146]

**Table 6 biomimetics-08-00205-t006:** Summarisation of materials, modification methods, intended tissue type, and which mechanical properties were altered and assessed during studies which utilised surface modification techniques.

Tissue Type	Materials	Fabrication Method	Modification Method	Mechanical Properties Modified and Assessed				Author, Year, Reference
				Tensile/ Compressive Modulus	Failure Strain	Ultimate Tensile strength	Storage/Loss Modulus	
Bone	PLA/HA/PDA	3D printing	PDA coating	✓				Chi et al. (2022) [160]
Bone	SF/OcPh/PDA	Freeze-drying	PDA coating	✓				Peng et al. (2022) [161]
Bone	Forsterite/Copper ferrite/P3HB	Sol–gel combustion	P3HB coating	✓				Aghajanian et al. (2022) [64]
Neural	PCL + Chitosan and PCL + Chitosan + Alginate	Electrospinning	Alginate coating	✓	✓	✓		Habibizadeh et al. (2022) [162]
Osteochondral	AC-dECM/ Bone-dECM/	Freeze-drying	Annealing	✓				Browe et al. (2022) [156]
Osteochondral	PCL/GelMA	PDMS mould UV curing	Salt leaching	✓				DiCerbo et al. (2022) [157]
Skin	PCL/PEG/ PCEC/Hydrogel	Electrospinning	Copolymer/hydrogel adsorption	✓			✓	Zhong et al. (2022) [163]
Skin	PCL/Chitosan/Gelatine	Electrospinning	Collagen grafting	✓	✓	✓		Sheikhi et al. (2022) [159]
Vascular	PCL/ ePTFE/RGD	Material/solvent solution	Induced crystallisation and RGD coating	✓	✓	✓		Wang et al. (2022) [164]
Vascular	PCL + PGS + DTβ4	Electrospinning and 3D printing	Molecular modification	✓	✓			Xiao et al. (2022) [165]

PDA: Polydopamine, OcPh: Octacalcium phosphate, P3HB: Poly-3-hydroxybutyrate, AC-dECM: articular cartilage decellurised extra-cellular matrix, ePTFE: expanded Polytetrafluoroethylene, RGD: Arginine-glycine-aspartate, PGS: Poly(glycerol sebacate), DTβ4: Dimeric thymosin β4, PEG: Poly(ethylene glycol), PCEC: PCL-b-PEG-b-PCL.

**Table 7 biomimetics-08-00205-t007:** An overview of Young’s modulus value ranges attained during studies presented in Table 6.

Tissue Type	Materials	Fabrication Method	Modification Method	Elastic Modulus Range Attained in Tension (T) or Compression (C)						Author, Year, Reference
				<1 kPa	1–100 kPa	100–1000 kPa	1–100 MPa	100–1000 MPa	>1 GPa	
Bone	PLA/HA/PDA	3D printing	PDA coating				✓ (C)	✓ (T)		Chi et al. (2022) [160]
Bone	SF/OcPh/PDA	Freeze-drying	PDA coating		✓ (C)					Peng et al. (2022) [161]
Bone	Forsterite/Copper ferrite/P3HB	Sol–gel combustion	P3HB coating				✓ (C)			Aghajanian et al. (2022) [64]
Neural	PCL + Chitosan and PCL + Chitosan + Alginate	Electrospinning	Alginate coating				✓ (T)			Habibizadeh et al. (2022) [162]
Osteochondral	PCL/ GelMA	PDMS mould UV curing	Salt leaching		✓ (C)	✓ (C)				Browe et al. (2022) [156]
Osteochondral	AC-dECM/Bone-dECM/	Freeze-drying	Annealing				✓ (C)			DiCerbo et al. (2022) [157]
Skin	PCL/PEG/ PCEC/Hydrogel	Electrospinning	Copolymer/hydrogel adsorption		✓ (C)					Zhong et al. (2022) [163]
Skin	PCL/Chitosan/Gelatine	Electrospinning	Collagen grafting				✓ (T)			Sheikhi et al. (2022) [159]
Vascular	PCL/ ePTFE/RGD	Material/solvent solution	Induced crystallisation and RGD coating				✓ (T)	✓ (T)		Wang et al. (2022) [164]
Vascular	PCL + PGS + DTβ4	Electrospinning and 3D printing	Molecular modification					✓ (C)		Xiao et al. (2022) [165]

**Table 8 biomimetics-08-00205-t008:** Synopsis of studies which utilised FEA analyses to model various tissues, including model type and whether the model was based on experimental or simulation data.

Tissue Type	Model Type	Based on Native Tissue Assessment?	Author, Year, Reference
Bone	Linear	✓	Irarrázaval et al. (2021) [182]
Cartilage	Non-linear	✗	Jahangir et al. (2022) [183]
Kidney	Linear	✗	Jing et al. (2021) [184]
Liver	Linear	✓	Fujimoto (2021) [185]
Neural	Non-linear	✓	Peshin (2023) [186]
Osteochondral	Non-linear	✗	Hislop et al. (2021) [187]
Skin	Linear	✓	Jobanputra et al. (2021) [188]
Vascular	Linear	✓	Helou et al. (2023) [189]

**Table 9 biomimetics-08-00205-t009:** Summary of materials and intended tissue type, as well as animal species study length, whether the proposed implant fully integrated into the animal, and a brief synopsis of all studies considered thus far, where animal testing was employed.

Tissue Type	Material Type	Fabrication Method	Translation Species	Length of Study	Minimised Immune Response?	Study Summary	Author, Year, Reference
Bone	Gelatine + HA and gelatine + HA + hPE	3D printing	Sprague Dawley rats	12 weeks	✓	The loading of hPE into the gelatine/HA scaffold induced a superior osteogenic response compared to that of the unmodified scaffold.	Lee et al. (2022) [117]
Bone	Col and Col + PGA and Col + PGA/HA + PGA and Col + PGA/HA + PGA (2-layer membrane)	3D printing	Sprague Dawley rats/Nude mice	4 weeks/1, 2, and 4 weeks	✓	The Col + PGA/HA + PGA scaffold indicated the highest cell proliferation and osteogenesis. The next highest cell viability was found in the Col + PGA scaffold. For the membrane-based scaffold, the cell core appeared on the surface of the membrane, with the ECM inside it. The Col scaffold showed the lowest cell viability.	Nguyen et al. (2022) [115]
Bone	PCL + Gel + HA and PCL + Gel + Hep/PCL + Gel + HA	Electrospinning and 3D printing	New Zealand white rabbits	5 and 20 weeks	✓	The composite scaffold showed good integrative and regenerative properties, and displayed no cytotoxicity, while also acting as a barrier to prevent infiltration by fibrous connective tissue. The bilayer scaffold demonstrated much greater new bone tissue formation, however.	Liu et al. (2021) [148]
Bone	Forsterite + copper ferrite and Forsterite + copper ferrite/P3HB	Sol–gel combustion	Wistar rats	8 weeks	✓	Both scaffolds induced a positive response in terms of new tissue formation and trabecular thickness when compared to the control study. However, the P3HB was observed to slightly increase these properties further. Additionally, neither specimen seemed to induce an immune response.	Aghajanian et al. (2022) [64]
Bone	Palm dECM + silicon (OTS-modified) and Palm dECM + silicon (APTES-modified)	Decellurizing	Wistar rats	2 and 4 weeks	✓	Neither scaffold presented signs of inflammation nor infection. Both scaffolds exhibited neovascularisation, the presence of endothelial cells, and collagen network fibres. The quantification of these differences was not presented.	Mahendiran et al. (2022) [166]
Cartilage	HA-NFs + GelMA	Hydrogel	Sprague Dawley rats	12 weeks	✓	Increasing quantities of HA-NFs in the GelMA promoted a stronger osteogenic response, with 15 and 25 wt/wt% HA-NFs showing new bone deposition and blood tissue formation, compared to 0 and 5 wt/wt% which showed little new tissue formation.	Wang et al. (2022) [118]
Cartilage	GelMA + HAMA + chondrospheroids	Hydrogel	Nude mice	1 and 2 months	✓	The chondro-spheroids maintained their morphology during the study. Genes COL 2, SOX 9, and HIF-1a were upregulated in comparison to the positive control (natural cartilage), while COL 10 was downregulated in comparison.	Wang et al. (2021) [113]
Cartilage	Alg + HA 50:50 (~70 layers) and Alg + HA 70:30 (~70 layers)	Bioprinting	C57BL/6 mice	1 and 4 weeks	✓	No significant differences were found, in terms of integration, between 50:50 and 70:30 ratios of Alg + HA; however, both scaffolds demonstrated high expression rates of macrophage F4/80 and angiogenesis protein CD31 compared to the control solution.	Janarthanan et al. (2022) [138]
Neural	PCL + Chitosan and PCL + Chitosan + Alginate	Electrospinning	Wistar rats	2, 4, and 8 weeks	✓	The PCL + Chitosan sheet showed a moderate inflammatory response and rapid degradation in comparison to the PCL + Chitosan + Alginate construct, which induced a mild inflammatory response and featured a slower degradation rate.	Habibizadeh et al. (2022) [162]
Osteochondral	Gellan gum + Alginate sodium and Gellan gum + Alginate sodium + TMP-BG	3D printing	New Zealand white rabbits	6 and 12 weeks	✓	After 6 weeks of implantation, subchondral bone growth was slightly diminished in the alginate + gellan growth, and significantly higher in the TMP-BG group, compared to the control. By week 12, the alginate + gellan group showed improved subchondral bone growth compared to the control, and the TMP group showed further enhanced proliferation.	Chen et al. (2022) [122]
Osteochondral	Col + PLGA and Ti/Col + PLGA	3D printing/freeze-drying	New Zealand white rabbits	4, 12, and 24 weeks	✓	For the Col-PLGA group, while it indicated superior cell proliferation at the defect site after 24 weeks, this was mostly just fibrous tissue. In comparison, the bilayered scaffold showed more new bone tissue and better integration with the host. The defect did not fully heal after 24 weeks for either construct.	Yang et al. (2021) [135]
Osteochondral	Ti/PLA/Col + PLGA and Col + HA	3D printing	Ovine condyle model	12 weeks	✓	The multi-layer scaffold provided a more homogenous response in terms of ‘filling in’ the defect. In addition, the Col + HA scaffold was rougher than the multi-layer, featuring cracks and fissures. In general, the multi-layer scaffold offered a more complete repair response.	Tamaddon et al. (2022) [139]
Osteochondral	PCL + PEO + PES and PCL + PEO/rGO + HAP-Sr + PES-BH_0.5%_	Electrospinning	Wistar rats	2 months	✓	The nanocomposite scaffold showed larger upregulation in the COL II, COL X, SOX 9, ALP, and Osteocalcin genes and protein compared to the hybrid scaffold. The hybrid scaffold may have induced an immune response due to degradation.	Dargoush et al. (2022) [134]
Osteochondral	AC-dECM/ Bone-dECM/	Freeze-drying	Caprine models	24 weeks	✓	Broad variation in defect repair quality was found. Generally, however, the bilayered scaffold promoted zonally defined tissue, and was able to return the mechanical properties of the region close to that of the surrounding osseous region. Bone repair was more consistent than that of the natural healing process.	Browe et al. (2022) [156]
Osteochondral	Silk and BMP-2 + silk and silk/silk (bilayer) and BMP-2 + bilayer and BMP-2 + bilayer/TGF-β1 + SilMA	Hydrogel solution	New Zealand white rabbits	0, 3, and 8 weeks	✓	The silk and BMP-2 + silk integrated poorly; however, the latter of these did promote large volumes of new bone tissue. Comparatively, the bilayer scaffold alone showed very little new tissue formation. The BMP-2 + bilayer integrated and promoted new tissue growth well, superseded only by the silk + SilMA composite scaffold.	Wu et al. (2021) [133]
Osteochondral	GelMA + PEO + HA/GelMA + PEO + HA/GelMA + PEO	Hydrogel solution	New Zealand white rabbits	12 weeks	✓	It was stated and illustrated that the tri-layer scaffold demonstrated good capacity for regenerating cartilage, subchondral bone, and trabecular bone. These properties were not quantified or examined further, however.	Li et al. (2022) [141]
Skin	PCL+ Col and PCL + Col + ZnO and PCL + Col + ZnO + VEGF	Electrospinning	Sprague Dawley rats	6 and 12 days	✓	Gross imaging and linked diagrams indicated that the wound healing rate was enhanced by the use of PCL + Col, and further enhanced by the inclusion of ZnO, before reaching its highest rate with the inclusion of Zno + VEGF. A similar trend was noted during Col and TGF-β1 expression analysis.	Li et al. (2021) [126]
Vascular	PCL + PGS + DTβ4	Electrospinning and 3D printing	New Zealand white rabbits	2 and 12 weeks	✗	Patency rate for scaffold was maintained at 80% across test animals, showing no signs of dilation or thrombosis. However, the grafts degraded before native tissue could remodel around the grafts. The slight generation of cross-linked elastin was noted, as well as rapid endothelialisation.	Xiao et al. (2022) [165]

**Table 10 biomimetics-08-00205-t010:** An overview of which organ-on-chip models are currently in use, illustrating from this subset which of these studies have considered the mechanical properties of the tissue in question.

Cell/Molecule Type	Chip Material Type/System	Fabrication Method	Study Purpose	Mechanical Properties Considered?	Author, Year, Reference
RAW264.7 macrophages and NIH-3T3 fibroblasts	Polydimethylsiloxane (PDMS)	Soft lithography	Skin wound healing	✗	Li et al. (2023) [195]
Chinese hamster ovary cells	Silicon	Micro-electro-mechanical system	Single cell analysis	✗	Xu et al. (2023) [196]
Caco-2 and HepG2 cells	PDMS	Soft lithography	Disease modelling (non-alcoholic fatty liver disease)	✗	Yang et al. (2023) [197]
Keratinocytes	PDMS	Soft lithography	Skin reconstruction	✗	Ahn et al. (2023) [198]
Hepatocytes, stellate cells, Kupffer-like macrophages, and endothelial cells	PDMS	Soft lithography	Studying effects of inflammation and cirrhosis on drug metabolism during hepatocellular carcinoma	✓	Özkan et al. (2023) [199]
Cardiomyocytes and cardiac fibroblasts	PDMS	Casting	Examining effects on cardiac tissue mechanical response following infarction	✓	Das et al. (2022) [200]
Glucose molecules	Polymethyl methacrylate	Stereolithography	Glucose sensing	✗	Podunavac et al. (2023) [201]
H9c2 rat cardiac myoblasts and adult human dermal fibroblasts	Agarose	Casting	Cardiac remodelling following arteriovenous fistula	✗	Waldrop et al. (2023) [202]
Human embryonic kidney 293 cells, NIH3T3 embryonic mouse fibroblasts, and human mammary MCF10A cells	PDMS	Soft lithography	Cell spheroid viscoelasticity quantification	✓	Boot et al. (2023) [203]
Human umbilical vein endothelial cells	PDMS	Stereolithography	Effect of biomechanical/biochemical stimuli on angio- and vasculogenesis	✗	Ferrari et al. (2023) [204]
Mouse podocytes and mouse glomerular endothelial cells	PDMS	Casting	Investigating crosstalk between glomerular endothelial cells and glomerular epithelial cells	✗	Hart et al. (2023) [205]
Taste receptor cells	MEA2100-System	Unstated	Ex vivo sense of taste simulation	✗	Wu et al. (2023) [206]
Bone marrow mesenchymal stem cells	PDMS	Casting	Cross-cellular interactions in osseous tissue	✓	Erbay et al. (2022) [207]
Human lung fibroblasts and human umbilical vein endothelial cells	PDMS	Casting	Controlling angiogenesis in lung cancer spheroids	✗	Kim et al. (2022) [208]
Bovine aortic endothelial cells and human pulmonary artery endothelial cells	PDMS	Spin coating and dual-layer lithography	Cell shear stress analysis and imaging	✓	Sinclair et al. (2023) [209]

**Table 11 biomimetics-08-00205-t011:** Summary of clinical trials with results published within the last two years, including material type, fabrication method and intended tissue type, procedure undergone by patient(s), length of subsequent study, post-operative outcome of patient(s), and a brief summary of each study.

Tissue Type (Region)	Material Type	Fabrication Method	Procedure	Study Length	Post-Operative Positive Outcome (Patient Number/Total Cases)	Study Summary	Author, Year, Reference
Bone (hand)	PLA	3D printing	Splint fitting	4–6 weeks	10/10	Compared to the control (thermoplastic) splint, patients reported a similar level of comfort, with the cost-effectiveness of the design potentially outperforming current designs. Two patients reported splint breakage after heavy use.	Waldburger et al. (2021) [212]
Bone (jaw)	PCL + β-TCP	3D printing	Maxillary reconstruction	6 months	8/8	While one patient suffered post-operative wound dehiscence (which was subsequently covered with a local flap), all cases showed signs of bone regeneration and scaffold integration. However, the group concluded that definitive parameters, such as implant efficacy and degradation, could not be accurately assessed.	Jeong et al. (2022) [31]
Bone (leg)	PCL + TCP	3D printing	Scaffold implantation	9–23 months	4/4	The scaffold, designed as a mesh which wraps around the bone defect site, was tested in four cases. In all cases, load-bearing functionality of the affected bone area was eventually restored, with bone development into the scaffold noted in three of the four cases.	Laubach et al. (2022) [213]
Bone (skull)	Surgical guide resin	3D printing	Cranioplasty	3 days	0/1	Patient mobility and cognition improved during the period of study. Patient symptoms began to redevelop, but not to the same extent as prior to the procedure, before passing away due to complications as a result of their initial condition.	Mee et al. (2021) [214]
Bone (skull)	PMMA	3D printing	Cranioplasty	10 days	3/3	The first case initially showed signs of fluid collection between the implant and dura; however, this was resolved shortly thereafter. The other two cases progressed well as expected.	Dabadi et al. (2021) [215]
Bone (sternum)	Aluminium oxide	Unspecified—company trademarked	Sternal replacement	28 months	1/1	The patient did not present any physiological complaints at the 28 month follow-up appointment, and was able to take part in physical activities without displacement or irritation of the prosthesis. This particular construct has a history of prior use with paediatric patients.	Mainard et al. (2022) [216]
Cardiac	UPy	Supramolecular chemistry	RVOT reconstruction	12 months	12/18	Overall, the conduit performed well across patients; however, several outlier complicated cases developed. These were either a result of immune response and thrombi formation, failure of the valve leaflets, or poor integration times. These were resolved using alternative measures.	Morales et al. (2021) [217]
Dental	Undefined resin—company trademarked	3D printing	Aligner fitting	76 weeks	120/120	Patients presented dental problems such as crowding, over-, under-, and cross-bite. All 120 cases were corrected using 3D-printed clear aligners and resulted in well-aligned teeth.	Yu et al. (2022) [218]
Dermal (face)	Unspecified thermoplastic	3D printing	Volumetric modulated arc therapy	12 weeks	23/35	Twelve of the patients treated with the 3D-printed bolus suffered from grade ≥2 radiation dermatitis, while seventeen did with the conventional bolus. Additionally, none of the patients suffered from radiation pneumonitis, which is significant compared to the three which did using the conventional bolus.	Zhang et al. (2021) [219]
Oral tissue	Acrylic	CAD/Casting	Impression of cleft palate tray	Procedure length (~16 min)	Not stated	In comparison to the standard ‘finger technique’ of the impression of cleft palate trays, the 3D-modelled acrylic tray recorded greater detail, showed zero distortions, took less time to obtain the impression, and induced a lower heart rate in the infants who were tested.	Kalaskar et al. (2021) [220]
Oral tissue/Limb	PLA, PLU-branded elastomer, TPU	3D printing	Splint fitting	-	1/18	Three sets of constructs were tested: oral, upper-extremity, and lower-extremity splints. The oral splints could not be tested fully as patients resisted their use. Only one patient tolerated the upper-extremity splint, which did remove the risk of contracture during its use. The lower-extremity splints caused discomfort due to tightness and incongruity. The group recommend the usage of dynamic splints as a remedial measure.	Şenayli et al. (2021) [221]
Renal	PLA	3D printing	Renal autotransplantation	12 months	1/1	This construct took the form of a 3D-printed cold jacket, for use during kidney transplantation. One year after the operation, an ultrasound scan indicated normal kidney size and shape, and that ordinary blood flow had resumed. However, random comparison to existing methods and/or extending this procedure to other patients were not considered.	Cui et al. (2022) [222]
Soft tissue	dECM	Cell sheet	Flap reconstruction	6 months	7/9	Integration of the latter cases was complicated by wound dehiscence. This was later resolved via secondary healing. Additionally, five patients received incisional negative pressure wound therapy, but the impact of which could not be fully assessed.	Desvigne et al. (2021) [223]

PMMA: Poly(methyl methacrylate), UPy; Supramolecular 2-ureido-4[1H]-pyrimidone, PLU: Polyurethane, TPU: thermoplastic polyurethane, TCP: Tricalcium phosphate, β-TCP: β-Tricalcium phosphate.

## Data Availability

The authors confirm that the data supporting the findings of this study are available within the article and its Appendix A.

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
