# Peer review of "Recent Methods for Modifying Mechanical Properties of Tissue-Engineered Scaffolds for Clinical Applications"

_biomimetics, 2023, doi:10.3390/biomimetics8020205_

Round 1
Reviewer 1 Report
I am recommending the manuscript for MAJOR REVISION, the comments given as follows:
1. Since the present review discuss about mechanical properties, needs to know to the different biomaterials have their characteristic, starting from metals, polymers, and ceramics that makes biomaterials selection is a vital issue. Please deliver this important topic and adopt the literature suggested, doi: 10.3390/su142013413
2. What makes the author's novelty in the present article? My analysis suggests that other similar previous articles properly explain the points you have brought up in the current paper. Please be sure to emphasize anything truly novel in this work in the introductory section.
3. Previous work must be explained in the introductory part, including their work, innovation, and limits, to demonstrate the literature gaps that will be filled in the current literature.
4. The reviewer encouraged the authors to provide an additional figure in the introduction section to improve the reader's understanding.
5. Line 56 the authors first time mentioning scaffold. The authors should explain the role and basic concept of scaffold in tissue engineering in brief. For this purpose, refer the relevant reference as follows: doi: 10.3390/su15010823 and 10.3390/biomedicines11020427
6. Line 301 the authors state of young’s modulus in previous study. This parameter describes input parameter used to describe the mechanical properties of hard materials during the computational simulation process. Please add additional information along with the reference as follows: doi: 10.3390/biomedicines11030951
7. Five years back literature should be enriched into the reference. MDPI reference is strongly recommended.
8. The reviewer suggests to the authors for reducing their self-citation the reverence of the present work.
9. The manuscript needs to be proofread by the authors since it has grammatical and language issues.
10. Following the revision step, the authors must provide a graphical abstract.
Reviewer 2 Report
The review by Johnston and Callanan explores the methods and techniques by which scientists have altered the mechanical properties of tissue engineered scaffolds to get mechanical compatibility between the scaffold and the graft/implantation site. Overall, the review is interesting and well-organised. In particular, the Tables provided in the Review could be very helpful for the readers of Biomimetics. I recommend the Review for acceptance, however there are some minor points that could be addressed before publication:
· Page 3, lines 105-106: “The extent of this review is all relevant work published within the years of 2021 to 2022”. It is not clear why the authors selected specifically only the last 2 years. For instance, why the authors did not cover a larger timeframe, such as the last 5 years (2018-2023)?
· In all Tables, for completion, the authors should have included more tissues, both soft and hard, subjected to mechanical loading. For example, among the soft tissues, it would be critically important to include ligament, tendons and muscles (as the cardiac muscle) as they are subjected to mechanical (and cyclic) loads, while enamel would be a great candidate alongside bone to represent hard tissues. This would give a broader range for comparison and a more impactful review.
· Page 14, lines 290-297: can you add some references to these statements?
· Page 16, line 338: “(v/v = 1/1)” should be replaced with “1:1 ratio”
· On page 24, the authors mention the possibility of using organ-on-a-chip to provide biomimetic substrates and overcome limitations of current models. Can the authors add a Table mentioning which organ-on-a-chip models are currently in use?
Round 2
Reviewer 1 Report
Reviewers greatly appreciate the efforts that have been made by the author to improve the quality of their articles after peer review. I reread the author's manuscript and further reviewed the changes made along with the responses from previous reviewers' comments. Unfortunately, the authors failed to make some of the substantial improvements they should have made making this article not of decent quality with biased, not cutting-edge updates on the research topic outlined. In addition, the author also failed to address the previous reviewer's comments, especially on comments number 1, 2, 3, 5, and 6. Please refer the comments precise as listed. Thank you very much for the opportunity to read the author's current work.
Round 3
Reviewer 1 Report
-